# MegaFlow: Large-Scale Distributed Orchestration System for the Agentic Era

## Abstract

The rapid development of interactive and autonomous AI systems signals our entry into the agentic era. Training and evaluating agents on complex agentic tasks such as *software engineering* and *computer use* requires not only efficient model computation but also sophisticated infrastructure capable of coordinating vast agent-environment interactions. However, no open-source infrastructure can effectively support large-scale training and evaluation on such complex agentic tasks. To address this challenge, we present **MegaFlow**, a large-scale distributed orchestration system that enables efficient scheduling, resource allocation, and fine-grained task management for agent-environment workloads. MegaFlow abstracts agent training infrastructure into three independent services (*Model Service*, *Agent Service*, and *Environment Service*) that interact through unified interfaces, enabling independent scaling and flexible resource allocation across diverse agent-environment configurations. In our agent training deployments, MegaFlow successfully orchestrates tens of thousands of concurrent agent tasks while maintaining high system stability and achieving efficient resource utilization. By enabling such large-scale agent training, MegaFlow addresses a critical infrastructure gap in the emerging agentic AI landscape.

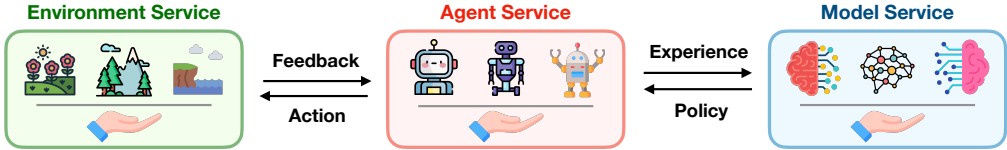

Figure 1: The proposed three-service architecture for agent training. **(left)** The *Environment Service* provides diverse interactive execution environments and returns feedback (observations, rewards, termination signals) in response to actions. **(middle)** The *Agent Service* orchestrates interaction, collects trajectories, and manages experiences. **(right)** The *Model Service* supports both inference (returning policies from context) and training (updating from experiences).

## 1 Introduction

The rapid development of interactive and autonomous AI systems signals our entry into the agentic era, where intelligent agents must be trained and evaluated on increasingly complex real-world tasks (Wang et al., 2024a; Xi et al., 2025). This transformation is driven by remarkable advances in large language models, reinforcement learning, and multi-agent coordination, enabling the development of agents capable of sophisticated reasoning and planning across diverse domains (Dorri et al., 2018). Training these agents on complex agentic tasks such as software engineering (Jimenez et al., 2023) and computer use (Xie et al., 2024) requires not only efficient model computation but also sophisticated infrastructure capable of orchestrating vast agent-environment interactions at unprecedented scale (Gao et al., 2024; Sun et al., 2025). The promise of large-scale agent training lies in its potential to develop more capable and versatile AI systems through massive parallel training across heterogeneous environments and tasks. Realizing this vision requires sophisticated infrastructure capable of supporting large-scale agent training and evaluation. However, no existing infrastructure

can effectively support the large-scale training and evaluation demands of such complex agentic tasks.

Traditional approaches to agent training work well for simple tasks such as single-turn function calling (Patil et al.) and basic question answering (Mialon et al., 2023). Nonetheless, they fail to address the unique challenges of orchestrating massive numbers of concurrent agent-environment interactions required for effective training on complex multi-step tasks at scale. The core challenge lies not merely in computational power (modern distributed computing frameworks have adequately addressed model training and inference scalability) but in the complex coordination of dynamic, interdependent processes that characterize large-scale agentic training workloads. Our experience training agents on complex tasks such as *software engineering* and *computer use automation* reveals three critical infrastructure bottlenecks that exemplify the broader scalability challenges facing large-scale agent training: (1) *Security and Isolation Constraints*: Complex agent training requires containerized environments to provide secure, isolated execution contexts for agent-environment interactions. However, security policies in typical training clusters prohibit the execution of arbitrary containers, creating a fundamental incompatibility between large-scale agent training requirements and existing computational infrastructure. (2) *Storage Scalability Limitations*: Each complex agent task instance requires corresponding containerized environments containing specific software dependencies and execution contexts. Even relatively modest datasets such as SWE-bench (Jimenez et al., 2023) and SWE-Gym (Pan et al., 2024) require over 25TB of storage for their associated container images. Storage requirements grow dramatically as training scales to larger and more diverse task sets, creating prohibitive infrastructure costs and management overhead. (3) *Computational Throughput Bottlenecks*: The resource-intensive nature of containerized agent-environment interactions severely limits concurrent training throughput, preventing the massive parallelism necessary for effective large-scale agent training.

To address these fundamental infrastructure challenges, we present **MegaFlow**, a large-scale distributed orchestration system that enables efficient scheduling, resource allocation, and fine-grained task management for agent training workloads. MegaFlow abstracts agent training infrastructure into three independent services (*Model Service*, *Agent Service*, and *Environment Service*) that interact through unified interfaces, enabling independent scaling and flexible resource allocation across diverse agent-environment configurations. The *Environment Service* provides diverse interactive execution environments and returns feedback (observations, rewards, termination signals) in response to actions. The *Agent Service* orchestrates interactions, collects experiences, and manages experience data throughout the training process. The *Model Service* supports both inference (returning policies from context) and training (updating model parameters from collected experiences). While existing approaches treat agent training as monolithic computational tasks, this modular architecture enables independent optimization and scaling of each component according to its specific computational requirements. The key insight underlying MegaFlow is that the primary scalability bottleneck in large-scale agent training lies not in model computation (which existing distributed frameworks handle well) but in the efficient coordination of dynamic agent-environment interactions. By providing unified APIs for the orchestration of these three services, MegaFlow enables researchers and practitioners to focus on algorithmic development rather than infrastructure complexity.

This work makes the following key contributions to large-scale agent training infrastructure:

- **Overcoming Security and Isolation Constraints:** We address the fundamental incompatibility between agent training requirements and cluster security policies by migrating containerized workloads to elastic cloud compute services. This enables secure, isolated agent execution without requiring specialized cluster configurations or compromising existing security frameworks.

- **Solving Storage Scalability Limitations:** We implement on-demand container image provisioning using cloud registry services with high-bandwidth internal network access, eliminating the need for massive local storage. This approach transforms storage requirements from a fixed infrastructure cost to an elastic, usage-based model that scales efficiently with training demands.

- **Breaking Computational Throughput Bottlenecks:** We introduce a distributed orchestration system that coordinates thousands of lightweight instances rather than relying on high-specification machines. Our many-small-instances approach achieves superior re-

source utilization and eliminates the availability constraints that limit traditional centralized methods to hundreds of concurrent tasks.

- **System Performance Validation:** We design and implement **MegaFlow**, a three-service architecture that enables independent scaling of model serving, agent coordination, and environment provisioning. Our evaluation demonstrates 32% cost reduction and consistent scaling to tens of thousands of concurrent tasks, with production validation across over 2 million agent training executions.

## 2 MEGAFLOW

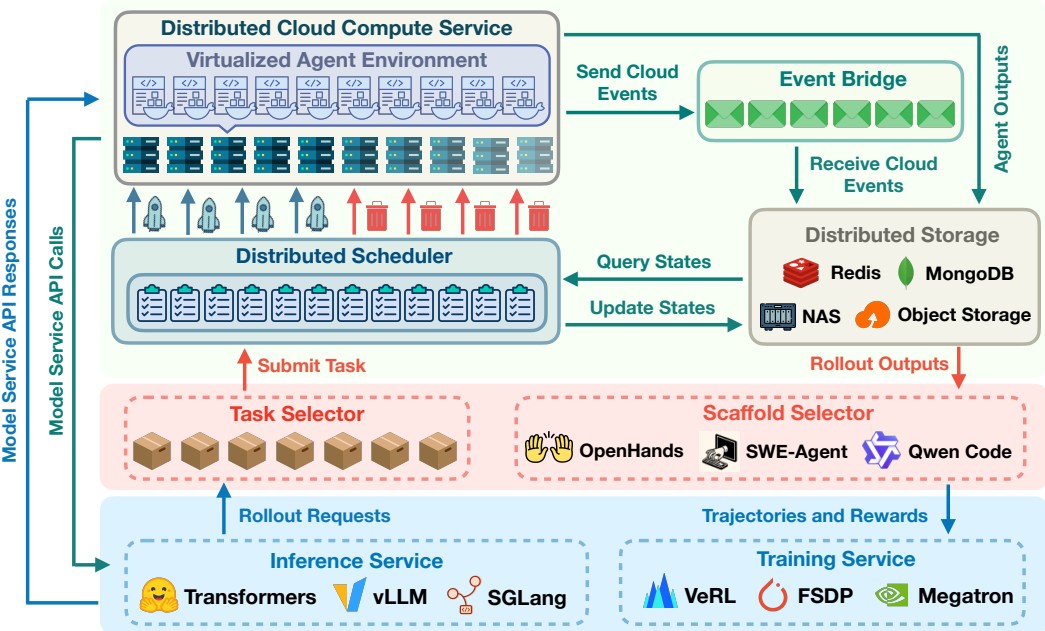

Figure 2: The architecture of **MegaFlow**. **(bottom)** The *Model Service* provides inference and training capabilities through various engines and distributed frameworks. **(middle)** The *Agent Service* coordinates execution strategies, integrates with agent frameworks, and manages experience feedback loops. **(top)** The *Environment Service* provides containerized execution environments and handles distributed task scheduling.

### 2.1 SYSTEM OVERVIEW

Figure 2 illustrates the overall architecture of MegaFlow's three-service ecosystem. The system operates through three independent services, each with specialized responsibilities:

**Model Service** This service handles the computational aspects of agent intelligence, providing inference capabilities through various inference engines such as Transformers (Wolf et al., 2020), vLLM (Kwon et al., 2023), and SGLang (Zheng et al., 2024), while supporting training operations via distributed training frameworks including VeRL (Sheng et al., 2025), FSDP (Zhao et al., 2023), and Megatron (Shoeybi et al., 2019). It focuses purely on model computation and parameter updates, abstracting away the complexities of agent-environment interactions.

**Agent Service** This service acts as the intelligent coordinator that manages agent execution strategies based on task requirements. It integrates with various agent frameworks such as Open-Hands (Wang et al., 2024b), SWE-Agent (Yang et al., 2024a), and Qwen Code (Yang et al., 2025a) for different task types (training, evaluation, or data synthesis) and coordinates rollout execution across specified datasets. The *Agent Service* processes rollout outputs, aggregating evaluation metrics and feeding experience data back to the *Model Service* for training iterations.

**Environment Service**    This service represents the most resource-intensive component, responsible for the physical execution of agent tasks. It queues tasks in a distributed system and employs sophisticated scheduling to monitor resource availability and dispatch tasks to cloud compute instances. Each instance executes multiple concurrent agent tasks through containerized environments that provide isolated execution contexts for agent-environment interactions.

**MegaFlow Orchestration**    MegaFlow orchestrates the interaction between these three services through unified APIs. It manages the complete lifecycle of agent training: from receiving requests and provisioning environments, to monitoring progress through event-driven updates, and collecting results for downstream processing. The system leverages cloud-native services for elastic compute, real-time monitoring, and distributed storage. While our current implementation is built on *Alibaba Cloud*, the abstracted APIs enable straightforward migration to other major cloud providers such as *Amazon Web Services*, *Microsoft Azure*, and *Google Cloud Platform*.

This architecture enables: (1) *elastic scaling* through dynamic resource allocation, (2) *fault tolerance* via event-driven monitoring, (3) *resource efficiency* through intelligent scheduling, and (4) *service isolation* allowing independent optimization of each component.

## 2.2    KEY DESIGN PRINCIPLES

The design of MegaFlow is guided by four key principles that reflect our understanding of large-scale agent training requirements and distinguish our approach from traditional distributed systems.

**Elastic Resource Strategy**    MegaFlow adopts a many-small-instances approach with standardized compute configurations, providing superior elasticity and cost optimization compared to few-large-instances models. This design aligns with containerized agent workload characteristics and enables rapid resource provisioning and deallocation.

**Hybrid Execution Model**    The system implements dual execution modes: *ephemeral* execution for perfect task isolation and *persistent* execution for resource efficiency. This hybrid approach optimizes both reliability and resource utilization based on task characteristics.

**Event-Driven Coordination**    Rather than complex consensus protocols, MegaFlow employs event-driven coordination with distributed state management, eliminating polling overhead while providing strong consistency guarantees for resource allocation and task scheduling.

**Specialized Component Delegation**    MegaFlow strategically delegates domain-specific operations to specialized systems (agent frameworks for container orchestration, cloud services for storage and monitoring), focusing on the unique challenges of agent-environment coordination rather than reimplementing general-purpose solutions.

## 2.3    ARCHITECTURE DESIGN

Based on these design principles, the MegaFlow architecture implements five core components that work in concert to provide scalable, fault-tolerant orchestration of agent training workloads.

**Task Scheduler**    At the heart of MegaFlow lies a high-performance asynchronous scheduler that enables massive concurrency for task processing. The system implements a FIFO scheduling policy, which proves sufficient for our workloads while maintaining simplicity and predictability.

The scheduler intelligently handles two distinct task categories with optimized resource allocation strategies. For *Ephemeral Tasks*, the system follows an ephemeral compute model: upon receiving a task request, a dedicated compute instance is provisioned, executes the single task, and is immediately deallocated, eliminating resource contention and providing perfect isolation. For *Persistent Tasks*, which require sustained execution, the scheduler maintains a pool of persistent compute instances and employs pool-based allocation, efficiently reusing resources while maintaining isolation through containerization.

**Resource Manager**    The resource management subsystem employs distributed coordination mechanisms to maintain real-time visibility into system state and resource availability. Rather than implementing complex resource monitoring and allocation algorithms, our design adopts a uniform

resource allocation strategy with standardized compute instances. This standardization simplifies scheduling decisions, improves resource predictability, and aligns with containerized workload characteristics where each instance typically executes a single agent task.

The system implements sophisticated concurrency control through a three-tier limiting mechanism: (1) User-specified parameters control the rate of *Model Service* API calls, preventing downstream bottlenecks; (2) Distributed semaphores ensure that task execution never exceeds available compute capacity; and (3) Administrative quotas provide control over resource usage, preventing system abuse while enabling fair resource sharing.

**Environment Manager**   Our environment management strategy demonstrates a key architectural insight: by delegating container lifecycle operations to proven open-source agent frameworks, MegaFlow focuses on what it does best (orchestration and coordination). The system pre-provisions all required container images in cloud registry services, enabling rapid deployment through high-bandwidth internal network access.

Environment isolation is achieved through a layered approach: each compute instance provides resource isolation, while containerization within instances provides process and filesystem isolation. This dual-layer isolation ensures that agent operations (including code editing, command execution, and file system modifications) remain completely contained within their designated environments.

**Event-Driven Monitoring**   MegaFlow employs cloud event services to implement reactive system behavior through two critical event streams. Instance lifecycle events enable the system to track compute instance state transitions, ensuring tasks are only dispatched to fully operational instances. Task completion events provide real-time notification of task outcomes, enabling immediate resource reclamation and result processing.

This event-driven architecture eliminates the need for expensive polling operations while providing near-instantaneous response to state changes. The system supplements event notifications with direct API calls for detailed task execution information, striking an optimal balance between real-time responsiveness and comprehensive monitoring.

**Data Persistence**   The system architecture separates concerns between operational data and result artifacts through specialized storage systems. Operational metadata (including task specifications, execution state, and compute instance information) is managed through document databases with schema validation and type safety. Task queues are implemented using in-memory storage systems, leveraging high-performance operations for rapid task dispatch.

Agent execution artifacts are persisted to cloud object storage, providing durable, scalable storage for trajectory data, evaluation results, and training artifacts. This separation allows the Agent Service to retrieve results asynchronously while maintaining system responsiveness during peak execution periods.

## 3  EVALUATION

We evaluate MegaFlow's performance on large-scale complex agent training tasks, focusing on multi-step software engineering scenarios that require containerized environments and sustained agent-environment interactions. These tasks present unique infrastructure challenges due to their need for sophisticated orchestration at concurrent execution scale. Since no existing infrastructure provides comparable functionality for such agent training orchestration, our evaluation compares MegaFlow against traditional high-specification centralized approaches and analyzes system performance characteristics.

### 3.1  EXPERIMENTAL SETUP

**Task Definition and Datasets**   We evaluate MegaFlow using software engineering agent training tasks that require containerized environments and sustained agent-environment interactions. Our evaluation leverages large-scale software engineering datasets (Jimenez et al., 2023; Pan et al., 2024; Zhang et al., 2025a; Yang et al., 2025b; Zan et al., 2025; Zhang et al., 2025b), conducting experiments with workloads scaling up to tens of thousands of concurrent tasks to demonstrate system performance at scale.

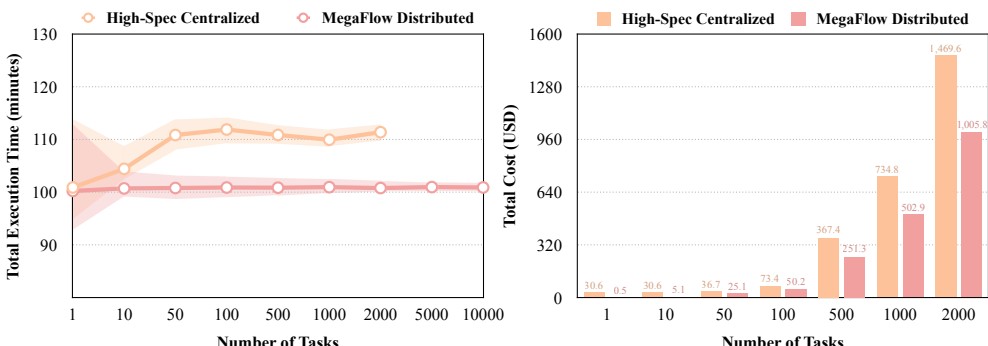

Figure 3: Throughput scaling and cost comparison between MegaFlow and centralized approaches. **(Left)** Total execution time showing MegaFlow's consistent performance versus centralized degradation. **(Right)** Total cost comparison with 32% cost reduction at 2,000 concurrent tasks. Data represents bootstrap sampling from over 130,000 production records.

**Agent Framework Compatibility** MegaFlow supports major agent frameworks including SWE-Agent (Yang et al., 2024a), OpenHands (Wang et al., 2024b), Mini-SWE-Agent (Yang et al., 2024a), Qwen Code (Yang et al., 2025a), and Claude Code (Anthropic, 2025) across all evaluated benchmark suites, validating our architecture's generalizability and broad compatibility with existing tools.

**Baseline Configurations** Since no comparable infrastructure exists for large-scale agent training orchestration, we establish baselines through systematic comparison of execution strategies:

- **High-Spec Centralized:** High-specification machines (208-core CPU, 3TB memory, 1 Gbps network bandwidth) with maximum sustainable parallelism of 50 concurrent tasks per instance.

- **MegaFlow Distributed:** Standardized 8-core, 16GB instances (100 Mbps network bandwidth each) with dynamic elastic scaling, where each instance handles 1 concurrent task.

**Data Collection and Analysis** Our evaluation is based on production deployment records comprising over 130,000 ephemeral execution tasks and over 2 million persistent execution tasks. Experiments utilized up to 40 high-specification instances for centralized approaches and up to 10,000 standardized instances for distributed approaches. Performance metrics are computed using bootstrap sampling (100 iterations per data point) with 95% confidence intervals. All experiments were conducted on *Alibaba Cloud*. Unless otherwise stated, we use ecs.re6.52xlarge instances for high-specification centralized approaches and ecs.c8a.2xlarge, ecs.c8i.2xlarge instances for distributed approaches.

## 3.2 Throughput and Scalability Analysis

We evaluate MegaFlow's scalability by measuring system performance across varying workload sizes, examining both throughput and latency characteristics under different concurrency levels.

**Performance and Scalability** Figure 3 demonstrates MegaFlow's superior characteristics compared to traditional centralized approaches. MegaFlow maintains consistent execution times of approximately 100 minutes across 1 to 10,000 tasks, while high-specification centralized methods exhibit degradation from 100 to 110 minutes due to resource contention bottlenecks. Centralized approaches suffer from network bandwidth congestion during container image pulls and resource competition during initialization. MegaFlow's distributed architecture eliminates these bottlenecks by providing dedicated resources per task.

The centralized approach faces fundamental scalability constraints, limited to 2,000 concurrent tasks due to instance availability (40 high-specification instances maximum). MegaFlow's standardized instances enabled provisioning up to 10,000 instances, demonstrating superior elastic scaling capabilities.

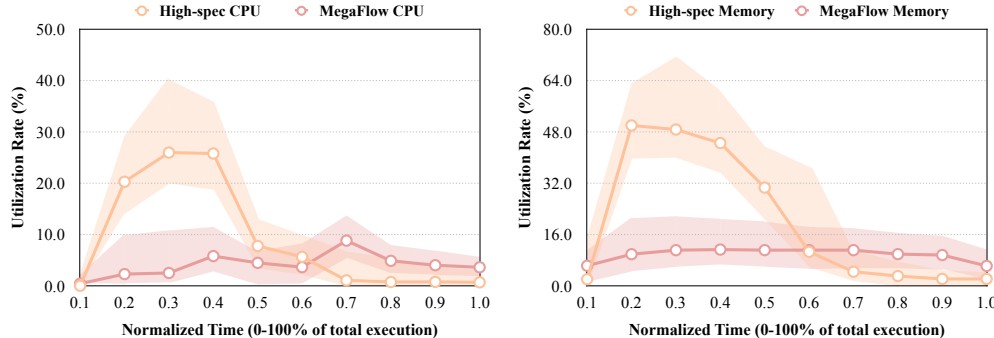

Figure 4: Resource utilization patterns across normalized execution time. **(Left)** CPU utilization: centralized peak at 25% versus MegaFlow's stable 5-10%. **(Right)** Memory utilization: centralized peak at 50% versus MegaFlow's consistent 12%. Shaded areas represent 95% confidence intervals.

**Cost Efficiency** At 2,000 tasks, MegaFlow achieves 32% cost reduction (1,005 vs 1,470 USD), with cost advantages increasing at larger scales. Beyond direct savings, MegaFlow eliminates resource availability constraints that prevent traditional methods from scaling to large workloads.

### 3.3 RESOURCE UTILIZATION ANALYSIS

We analyze resource utilization patterns throughout task execution lifecycles to evaluate the efficiency of different architectural approaches. Figure 4 presents CPU and memory utilization rates across normalized execution time for both approaches.

**Utilization Pattern Analysis** The resource utilization patterns reveal fundamental differences between centralized and distributed approaches. High-specification centralized instances exhibit pronounced resource usage spikes, with CPU utilization peaking at 25% during the initial 30% of execution time before declining to near-zero levels. Memory utilization follows a similar pattern, reaching 50% peak usage during mid-execution (20-40% of total time) then dropping sharply.

In contrast, MegaFlow's distributed architecture maintains consistent resource utilization throughout execution cycles. CPU utilization remains stable at 5-10% across the entire execution period, while memory utilization maintains approximately 12% with minimal variation.

**Resource Efficiency Implications** The contrasting utilization patterns highlight significant efficiency differences. Centralized approaches demonstrate typical "bursty" resource consumption with substantial idle periods, leading to poor overall resource efficiency despite high-specification hardware. The large confidence intervals in centralized approaches indicate high variability in resource demand, making capacity planning challenging.

MegaFlow's stable utilization patterns with narrow confidence intervals demonstrate predictable resource consumption, enabling more efficient capacity planning and resource allocation. While individual instances operate at lower peak utilization rates, the distributed model achieves better overall resource efficiency through consistent utilization across the execution lifecycle.

### 3.4 END-TO-END LATENCY ANALYSIS

We analyze the complete task execution pipeline to identify performance bottlenecks and validate our hybrid execution model design. Figure 5 presents latency breakdown across different execution phases and environment startup scaling characteristics.

**Latency Breakdown Analysis** The latency decomposition reveals significant differences in execution efficiency across approaches. MegaFlow's persistent execution mode achieves the lowest total latency at approximately 75 minutes, with minimal infrastructure overhead. The ephemeral execution mode requires approximately 90 minutes total, with additional environment startup costs, while high-specification centralized approaches exhibit the highest latency at 110 minutes due to resource contention across all execution phases.

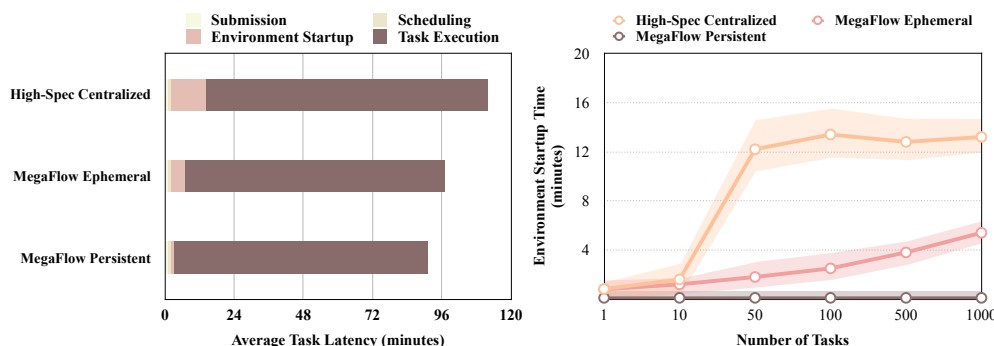

Figure 5: End-to-end latency breakdown and environment startup scaling comparison. **(Left)** Total execution times: MegaFlow Persistent (75 min), Ephemeral (90 min), and High-Spec Centralized (110 min). **(Right)** Environment startup time scaling showing centralized degradation (1-13 min) versus MegaFlow's stable performance.

Task execution represents the dominant component of total latency across all approaches, but infrastructure overheads vary substantially. Centralized approaches suffer from extended submission, scheduling, and environment startup phases due to resource competition and coordination bottlenecks.

**Environment Startup Scaling** Environment startup times demonstrate critical scalability differences between execution strategies. High-specification centralized approaches exhibit severe startup time degradation, increasing from 1 minute for single tasks to 13 minutes at 1,000 concurrent tasks due to resource contention during container image pulls and initialization. MegaFlow's ephemeral mode shows modest startup time growth from 1 to 6 minutes, while persistent execution maintains consistently low startup times below 1 minute across all concurrency levels through environment reuse.

The scaling patterns reveal multiple bottleneck sources. The modest increase in MegaFlow's ephemeral startup times suggests that cloud container registry services experience some performance degradation under high concurrent pull requests, but remain relatively stable. However, the dramatic startup time increase in centralized approaches indicates that the primary bottleneck lies in local resource constraints (network bandwidth limitations and resource contention within high-specification instances) rather than cloud service limitations. This analysis reinforces the effectiveness of MegaFlow's distributed approach in avoiding local resource bottlenecks through dedicated per-task resource allocation.

**Hybrid Execution Model Validation** These results validate our hybrid execution model design principle. Persistent execution provides optimal performance for sustained workloads through environment reuse, while ephemeral execution offers better isolation guarantees at moderate overhead. The ability to select execution modes based on task characteristics enables MegaFlow to optimize both performance and resource utilization according to specific workload requirements.

## 3.5 Discussion

Our evaluation demonstrates that MegaFlow successfully addresses the scalability challenges of large-scale agent training through distributed orchestration and hybrid execution models. The results validate our core design principles and provide several key insights for agent training infrastructure.

The many-small-instances approach achieves superior cost efficiency (32% reduction) while maintaining consistent performance, contrasting with centralized methods that suffer from resource contention bottlenecks. Resource utilization analysis reveals that stable, predictable consumption patterns enable more efficient capacity planning than bursty high-peak usage, challenging conventional assumptions about resource optimization in agent training systems.

Our bottleneck analysis identifies coordination overhead rather than raw computational power as the primary scalability constraint, with the distinction between cloud service limitations and local

resource constraints providing important design guidance. MegaFlow's broad compatibility across agent frameworks validates the practical value of infrastructure-level solutions for the research community.

Future work should explore orchestration of multi-environment agent tasks with complex service dependencies, potentially leveraging container orchestration paradigms like Kubernetes (Kubernetes Project, 2014) for dependency management. Additional directions include dynamic execution mode switching and multi-cloud deployment strategies to further enhance system flexibility.

## 4 RELATED WORK

Our work intersects several research areas, including distributed systems orchestration, containerization technologies, and infrastructure for AI workloads. We briefly review the most relevant prior work in each area, with additional comprehensive coverage provided in Appendix B.

**Distributed Container Orchestration**   Traditional container orchestration systems like Kubernetes (Kubernetes Project, 2014; Verma et al., 2015; Burns et al., 2016), Docker Swarm (Docker, 2025), and Apache Mesos (Mesos, 2025) focus on general-purpose workload management across distributed clusters. While these systems provide powerful abstractions for resource allocation and service discovery, they are not optimized for the unique characteristics of agent training workloads, such as rapid environment provisioning, heterogeneous execution requirements, and tight integration with model serving infrastructure.

**Cloud-Native AI Infrastructure**   Systems like Kubeflow (Kubeflow Project, 2018), MLflow (Zaharia et al., 2018; Chen et al., 2020), and Ray (Moritz et al., 2018) have advanced machine learning infrastructure by providing distributed training, model serving, and workflow orchestration capabilities. However, these systems primarily target traditional ML pipelines rather than interactive agent training workloads that require dynamic environment creation, containerized execution contexts, and complex agent-environment interaction patterns.

**Multi-Agent System Infrastructure**   Prior work on multi-agent systems has largely focused on coordination algorithms, communication protocols, and simulation environments (Sun et al., 2025). While recent efforts have explored infrastructure for LLM-based agents (Tran et al., 2025), these works primarily address single-agent scenarios or small-scale interactions. The infrastructure challenges of executing thousands of concurrent agent training tasks across distributed environments remain largely unaddressed.

**Large-Scale AI Training Systems**   Distributed training frameworks such as Horovod (Sergeev & Del Balso, 2018), FairScale (Facebook Research, 2020), and Megatron-LM (Shoeybi et al., 2019) have demonstrated the feasibility of coordinating AI workloads across large clusters. However, these systems focus on model training rather than agent execution, and their synchronous, tightly-coupled architectures are poorly suited to the asynchronous, loosely-coupled nature of agent-environment interactions.

Unlike existing approaches, MegaFlow provides a specialized three-service architecture that decouples model serving, agent coordination, and environment provisioning, enabling independent scaling and optimization for large-scale agent training workloads.

## 5 CONCLUSION

In this paper, we presented **MegaFlow**, a large-scale distributed orchestration system that addresses the fundamental scalability challenges facing agent training infrastructure through a three-service architecture that decouples *Model Service*, *Agent Service*, and *Environment Service*. Through comprehensive evaluation using over 130,000 production task records, we demonstrated that MegaFlow overcomes critical infrastructure bottlenecks, achieving 32% cost reduction and consistent performance scaling to 10,000 concurrent tasks compared to traditional centralized approaches. By establishing unified APIs and eliminating orchestration bottlenecks, MegaFlow provides a production-ready foundation for large-scale agent training research and enables the development of sophisticated AI agents at unprecedented scale.

## REPRODUCIBILITY STATEMENT

MegaFlow is built using standard cloud-native technologies and evaluated on *Alibaba Cloud* infrastructure. The architectural principles and implementation details provided in Section 2.3 are sufficient for independent implementation. We plan to release the system as open source to facilitate broader adoption and reproducibility.

## THE USE OF LARGE LANGUAGE MODELS

Large language models (LLMs) were used solely for language polishing and expression refinement to improve the clarity and readability of this paper. No LLMs were involved in the research design, system implementation, data analysis, or generation of technical content and conclusions.

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

# A  DEFINITIONS

This section provides formal definitions for the key concepts underlying MegaFlow's architecture and the agent training tasks it supports.

---

**Definition A.1: Three-Service Architecture**

An agentic system is decomposed into three modular services with well-defined interfaces:
**Model Service** ($\mathcal{M}$): Provides policy inference and training capabilities.

$$\text{Inference:} \quad \mathcal{M}_{\text{infer}} : \mathcal{S} \times \Theta \to \Pi(A) \tag{1}$$

$$\text{Training:} \quad \mathcal{M}_{\text{train}} : \mathcal{D} \times \Theta \to \Theta' \tag{2}$$

where $\mathcal{S}$ is the state space, $\Theta$ represents model parameters, $\Pi(A)$ is a policy distribution over actions, and $\mathcal{D}$ is training data.
**Agent Service** ($\mathcal{A}$): Manages agent coordination and experience collection.

$$\mathcal{A} : \mathcal{T} \times \mathcal{M} \to \mathcal{D} \times \mathcal{R} \tag{3}$$

where $\mathcal{T}$ is an agent task specification, and the output includes collected experiences $\mathcal{D}$ and execution results $\mathcal{R}$.
**Environment Service** ($\mathcal{E}$): Provides interactive environments and feedback signals.

$$\mathcal{E} : E_{\text{spec}} \times A \to \mathcal{S}' \times R \tag{4}$$

where $E_{\text{spec}}$ is the environment specification, $A$ is an action, and the output includes the next state $\mathcal{S}'$ and reward signal $R$.

---

To support large-scale execution of agent tasks across distributed infrastructure, MegaFlow decomposes the traditional monolithic agent system into three specialized services. This architectural separation enables independent scaling and optimization of each component while maintaining clear interfaces for coordination.

---

**Definition A.2: Agent Task**

An *Agent Task* is an interactive problem-solving environment defined as a six-tuple

$$\mathcal{T} = (E, D, G, \mathcal{S}, A, T),$$

where:

- $E$ is the environment specification that defines the interactive context in which the agent operates, including containerized execution environments, software repositories, or simulation parameters;

- $D$ is the task description that provides natural language instructions, problem statements, or objectives that guide the agent's behavior;

- $G$ is the goal and evaluation criteria that specify success conditions, reward functions, or metrics used to assess task completion and performance;

- $\mathcal{S}$ is the state space representing all possible configurations of the environment and agent context during task execution;

- $A$ is the action space defining the set of permissible operations the agent can perform within the environment;

- $T : \mathcal{S} \times A \to \mathcal{S}$ is the transition function that specifies how the task state evolves given an agent action $a \in A$, capturing the deterministic or stochastic dynamics of the environment.

An agent task execution produces a trajectory $\tau = \{(s_0, a_0), (s_1, a_1), \ldots, (s_T, a_T)\}$ where $s_t \in \mathcal{S}$ and $a_t \in A$. Task termination is controlled by the agent framework, either through explicit completion decisions or upon reaching the maximum step limit. During execution, the environment processes each action to generate state transitions, while the final reward signal $R = G(\tau)$ is computed only upon task completion using the complete trajectory.

---

Table 1: Compatibility matrix of agent frameworks with software engineering datasets

| Dataset | SWE-Agent | OpenHands | Mini-SWE-Agent | Qwen Code | Claude Code |
|---|---|---|---|---|---|
| SWE-bench | ✔ | ✔ | ✔ | ✔ | ✔ |
| SWE-Gym | ✔ | ✔ | ✔ | ✔ | ✔ |
| SWE-rebench | ✔ | ✔ | ✔ | ✔ | ✔ |
| SWE-smith | ✔ | ✔ | ✔ | ✔ | ✔ |
| SWE-bench Multilingual | ✔ | ✔ | ✔ | ✔ | ✔ |
| Multi-SWE-Bench | ✔ | ✔ | ✔ | ✔ | ✔ |
| Multi-SWE-RL | ✔ | ✔ | ✔ | ✔ | ✔ |
| SWE-bench-live | ✔ | ✔ | ✔ | ✔ | ✔ |
| SWE-Flow | ✔ | ✔ | ✔ | ✔ | ✔ |
| Terminal-bench | ✔ | ✔ | ✔ | ✔ | ✔ |

## B    ADDITIONAL RELATED WORK

This section provides comprehensive coverage of related work in software engineering agent systems, computer use automation, and benchmark datasets that complement the core infrastructure focus of our main paper.

**Software Engineering Agent Systems**    The field of software engineering automation has seen significant advancement with the development of specialized agent frameworks. SWE-Agent (Yang et al., 2024a) pioneered the application of language models to software engineering tasks through interactive code editing and testing. OpenHands (Wang et al., 2024b) provides a comprehensive platform for building generalist AI agents that can write code, interact with command lines, and browse the web, with support for multi-agent coordination and extensive evaluation benchmarks. More recent efforts include Qwen-Agent (Qwen Team, 2025), which focuses on Chinese language support, and specialized frameworks like Cursor and GitHub Copilot that target specific development workflows. While these frameworks demonstrate the potential of AI-assisted software development, they primarily operate at small scales and lack the infrastructure support for large-scale distributed training and evaluation.

**Software Engineering Benchmarks and Datasets**    The development of standardized benchmarks has been crucial for advancing software engineering AI. SWE-bench (Jimenez et al., 2023) established the foundation with real-world GitHub issues and corresponding fixes. SWE-gym (Pan et al., 2024) extended this with interactive environments for iterative development. Recent efforts have expanded coverage with SWE-bench Multimodal (Yang et al., 2024b), SWE-bench Multilingual (Yang et al., 2025b), Multi-SWE-bench (Zan et al., 2025), and language-specific variants. Terminal-bench (Team, 2025) evaluates command-line interactions, while Tau-bench (Yao et al., 2024) focuses on tool use capabilities essential for software development workflows. These datasets provide the foundation for systematic evaluation but require sophisticated infrastructure for large-scale concurrent execution.

**Computer Use and Browser Automation**    Beyond software engineering, agent systems have expanded to general computer use automation. WebArena (Zhou et al., 2023) introduced realistic web-based task environments for agent evaluation. Mind2Web (Deng et al., 2023) focuses on web navigation and interaction tasks. OSWorld (Xie et al., 2024) provides comprehensive operating system interaction benchmarks. Recent work on computer use agents includes Anthropic's computer use capabilities and similar efforts from other organizations. Browser automation frameworks like Playwright and Selenium provide the underlying execution capabilities, but lack the orchestration infrastructure for large-scale agent training scenarios.

## C    AGENT FRAMEWORK COMPATIBILITY

Table 1 demonstrates MegaFlow's comprehensive compatibility across major agent frameworks and software engineering datasets. The system's unified API abstraction enables seamless integration

Table 2: Overview of the RL training environments before and after filtering. Instances with pass rate equal to 1 (very easy) or 0 (very difficult) are removed to stabilize large-scale rollouts.

| Dataset | Environments (Before) | Environments (After) |
|---|---|---|
| SWE-Gym | 2,438 | 1,219 |
| SWE-rebench | 21,336 | 6,390 |
| Multi-SWE-RL | 4,723 | 924 |
| Synthesized (Internal) | 30,274 | 15,017 |
| **Total** | **58,771** | **23,550** |

with diverse agent implementations, including SWE-Agent (Yang et al., 2024a), OpenHands (Wang et al., 2024b), Mini-SWE-Agent (Yang et al., 2024a), Qwen Code (Qwen Team, 2025), and Claude Code (Anthropic, 2025), across all evaluated benchmark suites.

This broad compatibility validates MegaFlow's architectural design principle of specialized component delegation, where the system focuses on orchestration and coordination rather than reimplementing agent-specific logic. By abstracting infrastructure complexity behind standardized interfaces, MegaFlow enables researchers to leverage their preferred agent frameworks while benefiting from large-scale distributed execution capabilities.

## D  REINFORCEMENT LEARNING FOR AGENT TRAINING

In our reinforcement learning setup, we leverage MegaFlow to orchestrate large-scale Agentic Reinforcement Learning, enabling end-to-end training of coding agents directly within realistic software engineering environments. The system's distributed execution model allows us to run high-cost SWE environments (requiring real compilation, testing, and verification) at scale, a capability that conventional RL training infrastructures cannot provide.

### D.1  TRAINING SETUP

**Dataset**  The RL training corpus combines several publicly available SWE-style datasets, including *SWE-Gym* (Pan et al., 2024), *Multi-SWE-RL* (Zan et al., 2025), and *SWE-rebench* (Badertdinov et al., 2025) (using data released prior to March 2025), together with a large collection of internally synthesized SWE-style repair tasks. Before training, the combined corpus contains 2,438 environments from SWE-Gym, 21,336 from SWE-rebench, 4,723 from Multi-SWE-RL, and 30,274 synthesized environments. To stabilize large-scale RL rollouts, we filter out instances that are either very easy (pass rate equal to 1) or very difficult (pass rate equal to 0). Table 2 summarizes the environment counts before and after filtering. After filtering, approximately 1,219 SWE-Gym environments, 6,390 SWE-rebench environments, 924 Multi-SWE-RL environments, and 15,017 synthesized environments remain. All filtered environments are merged into a single unified rollout pool, and instances are sampled uniformly at random during RL training. Since sampling is performed without weighting or curriculum, the effective training distribution is directly determined by the post-filtering dataset sizes.

**Algorithm and Hyperparameters**  We train the agent using the *Group Sequence Policy Optimization (GSPO)* (Zheng et al., 2025) algorithm and integrate multiple agentic coding frameworks, including OpenHands (Wang et al., 2024b), SWE-agent (Yang et al., 2024a), mini-SWE-agent (Yang et al., 2024a), Qwen Code (Qwen Team, 2025), and Claude Code (Anthropic, 2025), ensuring broad generalization across heterogeneous agent designs. During RL training, MegaFlow orchestrates a total of 1024 parallel SWE environments for each training step. These environments correspond to 64 distinct SWE instances, and for each instance MegaFlow launches $n = 16$ independent rollout replicas. This hierarchical parallelism structure (64 tasks $\times$ 16 replicas) enables the agent to explore multiple trajectories per problem while sustaining extremely high throughput across heterogeneous software environments. We then apply GSPO for policy optimization using a minibatch size of 64 and 2 PPO epochs. The agent is allowed up to 100 interaction rounds per task, with a 128k-token context window enabling long-range reasoning over repository state, intermediate tool outputs, and

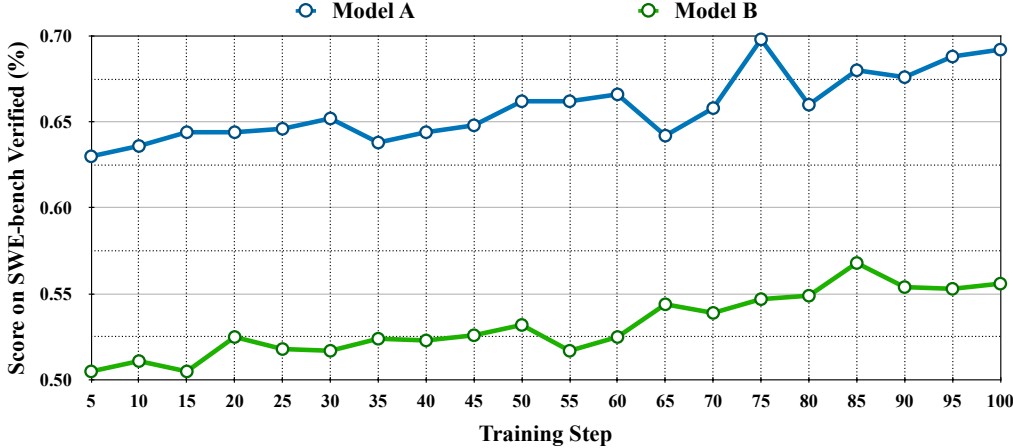

Figure 6: RL training dynamics on SWE-bench Verified. **Model A** is a 235 billion parameters MoE model and **Model B** is a 30 billion parameters MoE model. Scores are evaluated using the OpenHands scaffold across training steps 0–100.

prior attempts. The sampling temperature is set to 1.0, and the maximum response length per turn is limited to 4096 tokens. If the agent does not explicitly terminate the task within 100 rounds (i.e., does not call `finish`/`submit`), a fixed penalty of $-0.5$ is applied. We optimize the model using a learning rate of $1e-6$, with positive and negative reward clipping thresholds set to $4e-4$ and $2e-4$ respectively.

## D.2 TRAINING RESULTS

Figure 6 reports the RL training dynamics evaluated on SWE-bench Verified using the OpenHands scaffold. We compare two models of different scales: **Model A**, a 235 billion parameters MoE model, and **Model B**, a 30 billion parameters MoE model. Both models show consistent improvement during RL training, with the larger model achieving substantially higher scores throughout training. These results demonstrate the effectiveness of large-scale agentic RL training for SWE-style tasks and highlight MegaFlow's ability to sustain stable, fault-tolerant, and high-throughput distributed rollouts across heterogeneous agent frameworks.

