# OpenReview forum: "MegaFlow: Large-Scale Distributed Orchestration System for the Agentic Era"
_ICLR.cc/2026/Conference — Submitted to ICLR 2026_

### Official Review · Reviewer_YBFx · 2025-10-30

**Soundness:** 3
**Presentation:** 3
**Contribution:** 3
**Rating:** 6
**Confidence:** 1

**Summary:**

The paper introduces MegaFlow, which is a distributed orchestration system designed for training and evaluating agentic workloads. The key idea is a three-service decomposition: Environment, Agent, and Model. The authors demonstrate large-scale deployments with notable improvements.

**Strengths:**

I am not an expert in large-scale systems. My evaluation can only focus on clarity and methodological soundness. The three-service decomposition sounds clean and well-motivated to me, although I am not sure about the main literature in this field. The experimental results seem convincing, but I am not familiar with the setups. Since I am less familiar with this field, I recommend seeking opinions mainly from other reviewers with stronger expertise.

**Weaknesses:**

Since I am not an expert in this field, I found it difficult to grasp the key contributions of the paper. As a result, I was unable to provide meaningful weaknesses. I recommend seeking opinions from other reviewers with stronger expertise.

**Questions:**

Same as above.

---

> ### Author Response · Authors · 2025-11-23
> **Responses to Reviewer YBFx (1/n)**
>
> Dear reviewer,
>
> Thank you for taking the time to review our submission and for sharing your perspective. We appreciate your honesty regarding the limits of your expertise, and we will further refine the exposition in the revised version to make the key contributions clearer to a broader audience.
>
> Thank you again for your constructive evaluation.

---

### Official Review · Reviewer_zBgH · 2025-10-30

**Soundness:** 3
**Presentation:** 3
**Contribution:** 3
**Rating:** 6
**Confidence:** 2

**Summary:**

This paper presents MegaFlow, a large-scale distributed orchestration system tailored for training and evaluating autonomous agents in complex, interactive environments. The system decouples agent training into three modular services, with unified APIs and elastic cloud-based resource allocation. MegaFlow aims to overcome practical bottlenecks in agent-environment training pipelines, including security and isolation constraints, storage overhead from task-specific containerized environments, and throughput limitations of centralized high-spec clusters.

**Strengths:**

1. The paper articulates key system-level challenges in scaling interactive agent training, differentiating this setting from traditional large-model training workloads.
2. The three-service architecture is well-structured and clearly explained.

**Weaknesses:**

I am not an expert in agent orchestration, so please correct my mistakes in my questions:

1. Comparisons seem to be mainly against high-spec centralized machines rather than alternative distributed or hybrid systems.
2. While the modular three-service abstraction is intuitive, the paper would benefit from clearer articulation of which components introduce fundamentally new design ideas versus mature cloud-native practices adapted to the agent training context.
3. While the system enables large-scale rollouts, can authors offer analyses about whether this orchestration translates into better learning outcomes (e.g., improved agent capabilities, faster convergence) beyond execution efficiency?

**Questions:**

Please see the weakness.

---

> ### Author Response · Authors · 2025-11-23
> **Responses to Reviewer zBgH (1/n)**
>
> Dear reviewer,
>
> Thank you for taking the time to review our submission. Below, we provide point-by-point responses to your concerns and suggestions.
>
> ---
>
> > **Concern 1:** Comparisons seem to be mainly against high-spec centralized machines rather than alternative distributed or hybrid systems.
>
> SWE/CUA workloads differ substantially from conventional distributed or hybrid execution settings. Each task requires a fully isolated Docker/VM image, a complete repository environment, and multi-step interactions with persistent state. To the best of our knowledge, there is currently no existing open-source distributed framework that can orchestrate large numbers of such heavyweight, stateful SWE tasks end-to-end. Generic distributed systems (e.g., Ray, serverless compute platforms, or workflow schedulers) do not provide the strong environment isolation, per-instance dependency management, repository-level execution, or long-horizon control required for SWE/CUA rollouts. For this reason, high-spec centralized machines serve as the only meaningful baseline for comparison, capturing the best non-distributed alternative available for running these workloads.
>
> > **Concern 2:** While the modular three-service abstraction is intuitive, the paper would benefit from clearer articulation of which components introduce fundamentally new design ideas versus mature cloud-native practices adapted to the agent training context.
>
> Although MegaFlow builds on cloud primitives, no existing cloud platform provides the execution model required for large-scale agentic SWE/CUA workloads. Cloud services are fundamentally designed for stateless, short-lived, and homogeneous tasks (e.g., serverless functions, batch jobs, generic container orchestration). In contrast, SWE/CUA rollouts require per-task VM/Docker images, repository-level environments, multi-minute stateful execution, filesystem and dependency isolation, and reproducibility guarantees across millions of long-horizon agent interactions. These requirements fall entirely outside the capabilities of current cloud offerings. MegaFlow therefore does not replicate cloud functionality; rather, it supplies the missing systems layer that bridges cloud primitives with the execution semantics needed for real-world agent training at scale.
>
> > **Concern 3:** While the system enables large-scale rollouts, can authors offer analyses about whether this orchestration translates into better learning outcomes (e.g., improved agent capabilities, faster convergence) beyond execution efficiency?
>
> We appreciate the reviewer’s insightful question. In response to this suggestion, we have added Appendix Section D, which presents our agentic reinforcement learning experiments on MegaFlow. The learning curves included there (e.g., on SWE-Bench Verified) show a clear upward trajectory over the first 100 training steps, indicating that agents trained on MegaFlow do improve their capabilities over time. While the main contribution of the paper is the underlying systems infrastructure, these results demonstrate that MegaFlow can effectively support large-scale agent learning.

---

### Official Review · Reviewer_VxSd · 2025-10-30

**Soundness:** 3
**Presentation:** 3
**Contribution:** 2
**Rating:** 6
**Confidence:** 3

**Summary:**

This work introduces MegaFlow, a distributed system for efficiently connecting environments, agent scaffolds, and LLMs for training. This system solves many of the practical issues around agent training, like security constraints, storage space, and the need for powerful machines to run the environments, resulting in a significant cost reduction.

**Strengths:**

- This work addresses a significant challenge regarding scaling up data collection for agentic LLM training.
- The results and analysis are comprehensive from a systems perspective.
- The text is well-written and easy to follow.

**Weaknesses:**

- There are no results regarding the downstream utility of MegaFlow in the context of LLM training. I recognize that this is more of an infrastructure/systems paper, but any downstream results would've been appreciated.
- The CPU utilization and memory utilization is consistent but still low.

**Questions:**

- Does MegaFlow support conducting multiple concurrent tasks on each (8-core, 16 GB) system? Would this help increase utilization (at the cost of increasing latency)?
- MegaFlow seems to support a lot of existing LLM agent infrastructure, so I was wondering approximately how difficult would it be for a new user to start using MegaFlow (in terms of lines of boilerplate code or general user time)?

---

> ### Author Response · Authors · 2025-11-23
> **Responses to Reviewer VxSd (1/n)**
>
> Dear reviewer,
>
> Thank you for taking the time to review our submission. Below, we provide point-by-point responses to your concerns and suggestions.
>
> ---
>
> > **Concern 1:** There are no results regarding the downstream utility of MegaFlow in the context of LLM training. I recognize that this is more of an infrastructure/systems paper, but any downstream results would've been appreciated.
>
> We appreciate the reviewer’s point. While MegaFlow is primarily an infrastructure framework, we agree that illustrating its downstream utility is valuable. In response to reviewer feedback, we have added Appendix Section D, which presents our agentic reinforcement learning setup, including training two anonymized models (Model A and Model B). These results demonstrate that MegaFlow can support large-scale agent-based LLM training.
>
> > **Concern 2:** The CPU utilization and memory utilization is consistent but still low.
>
> We agree that the current CPU and memory utilization is relatively low, and this is an area we are actively optimizing. Since existing SWE datasets do not specify the compute resources required for successfully passing their test suites, we adopted a conservative instance configuration to ensure correctness across all tasks. Profiling shows that agent rollouts typically consume around 1.5 CPU cores and 2.3 GB of memory, while the evaluation stage averages roughly 2.5 CPU cores and 1.5 GB of memory, with some variation across tasks and agent frameworks. Based on these measurements, we are gradually reducing the instance size to avoid over-provisioning and to improve overall efficiency.
>
> > **Concern 3:** Does MegaFlow support conducting multiple concurrent tasks on each (8-core, 16 GB) system? Would this help increase utilization (at the cost of increasing latency)?
>
> MegaFlow is architecturally capable of executing multiple concurrent tasks on the same ECS instance. However, we intentionally enforce a one-task-per-instance execution model. SWE tasks exhibit highly variable CPU, memory, and network usage, and running multiple tasks on the same host introduces unpredictable resource contention that can affect correctness and reproducibility. Strong isolation is therefore a core design principle of MegaFlow.
>
> As noted in our response to the previous concern, our profiling shows that individual SWE rollouts and evaluations consume significantly less than the original 8-core, 16-GB allocation. Based on these measurements, we have already reduced the default instance type to ecs.c8i.xlarge and ecs.c8a.xlarge (4 cores, 8 GB). We also plan to explore finer-grained instance sizing based on per-framework profiling, although for simplicity and consistency, we currently use a uniform instance configuration across all tasks.
>
> > **Concern 4:** MegaFlow seems to support a lot of existing LLM agent infrastructure, so I was wondering approximately how difficult would it be for a new user to start using MegaFlow (in terms of lines of boilerplate code or general user time)?
>
> The effort required to start using MegaFlow is minimal. For users with an existing Kubernetes cluster, MegaFlow workflows are fully compatible with native Kubernetes job specifications, so no additional boilerplate is needed beyond submitting a standard workflow to the cluster. For users without a Kubernetes deployment, MegaFlow can also be accessed directly through cloud APIs, enabling rollout execution without any cluster setup.

---

### Official Review · Reviewer_vXD8 · 2025-11-01

**Soundness:** 2
**Presentation:** 2
**Contribution:** 2
**Rating:** 4
**Confidence:** 2

**Summary:**

The manuscript introduces MegaFlow, a distributed orchestration system for large-scale agent training, and claims cost and scaling advantages over centralized baselines.  Nevertheless, the decomposition into model/agent/environment services is straightforward, and the empirical comparison is confined to an internal cloud deployment without reproducible artifacts.

**Strengths:**

1. The three-service modularization yields a clean separation of concerns that simplifies independent scaling and maintenance.
2. The evaluation dataset is substantial (though I have no idea what the dataset essentially is), providing a degree of empirical credibility rarely seen in infrastructure proposals.

**Weaknesses:**

1. Figure 1 contains limited information; I suggest either reducing its space allocation or adding more explanatory details to enhance clarity.
2. In Line 215, what exactly are the “complex resource monitoring and allocation algorithms”? Likewise, what does the “standardized compute instance” implemented by the authors refer to? In Line 246, more details are needed regarding the document database—specifically, the structure of the operational metadata, its storage format, and how it is managed and retrieved. Without these details, it is difficult to discern the novel contribution of the proposed framework.
3. The evaluation setup is unclear. In Line 302, the authors mention “30,000 ephemeral execution tasks and over 2 million persistent execution tasks.” What exactly do these tasks represent, and how were they generated? Are they derived from specific training datasets?
4. Are there truly no comparable baselines? Could approaches such as VERL agent training or frameworks like AReal serve as baselines?
5. The authors claim this is an agent training framework, yet the experimental section lacks details on what LLM was trained, what data were used, and what hyperparameter configurations were applied, which makes the setup somewhat confusing.

**Questions:**

See Weakness

---

> ### Author Response · Authors · 2025-11-23
> **Responses to Reviewer vXD8 (1/n)**
>
> Dear reviewer,
>
> Thank you for taking the time to review our submission. Below, we provide point-by-point responses to your concerns and suggestions.
>
> ---
>
> > **Concern 1:** Figure 1 contains limited information; I suggest either reducing its space allocation or adding more explanatory details to enhance clarity.
>
> Figure 1 is intentionally presented as a high-level overview meant to convey the core idea of our three-service modular architecture at a glance, while the technical details are fully described in Section 2. Following the reviewer’s advice, we would slightly reduced its space allocation to improve the overall layout.
>
> > **Concern 2:** In Line 215, what exactly are the “complex resource monitoring and allocation algorithms”? Likewise, what does the “standardized compute instance” implemented by the authors refer to? In Line 246, more details are needed regarding the document database—specifically, the structure of the operational metadata, its storage format, and how it is managed and retrieved.
>
> - The phrase “complex resource monitoring and allocation algorithms” is intended only to contrast our straightforward scheduling logic—new tasks are executed whenever resources and quotas permit, and otherwise remain pending; although we rely on the cloud platform for resource provisioning, the actual scheduling and concurrency control are implemented by our system.
> - The term “standardized compute instance” refers specifically to a fixed VM configuration (8 vCPUs, 16 GB memory) mapped to Alibaba Cloud instance types such as ecs.c8a.2xlarge and ecs.c8i.2xlarge.
> - Regarding the document database, our operational metadata is managed in MongoDB through Beanie, which provides schema validation via Pydantic; it stores lightweight information such as user records, task specifications, execution states, and related bookkeeping. We will revise the corresponding sentences to explicitly state these meanings.
>
> > **Concern 3:** The evaluation setup is unclear. In Line 302, the authors mention “30,000 ephemeral execution tasks and over 2 million persistent execution tasks.” What exactly do these tasks represent, and how were they generated? Are they derived from specific training datasets?
>
> The “30,000 ephemeral execution tasks” and “over 2 million persistent execution tasks” correspond to real SWE tasks executed during the training and evaluation of our models. Each task consists of a full agent rollout on a SWE instance, followed by a verification step that checks whether the generated patch resolves the issue. These tasks originate from open-source SWE datasets such as SWE-bench, SWE-Gym, and SWE-rebench, as well as our internally synthesized SWE data. We will clarify this in the revision.
>
> > **Concern 4:** Are there truly no comparable baselines? Could approaches such as VERL agent training or frameworks like AReal serve as baselines?
>
> We appreciate the reviewer’s comment. However, VeRL and AReal operate in a fundamentally different problem setting from SWE and CUA. SWE/CUA tasks require per-instance Docker/VM images, full repository environments, persistent multi-step execution, and strict isolation across hundreds of thousands of agent rollouts. The dominant challenge is not “agent training,” but the systems bottleneck of provisioning and managing massive numbers of heavyweight environments with long-horizon interactions.
>
> VeRL does not address environment provisioning, runtime isolation, or large-scale orchestration, and therefore cannot serve as a baseline. Moreover, tightly coupling the agent implementation with the training framework (as VeRL does) becomes a severe limitation at scale: VeRL can sustain small-scale training where each step performs only thousands of rollouts, but it cannot feasibly support tens or hundreds of thousands of concurrent agent rollouts required by large-scale SWE/CUA RL. In contrast, MegaFlow’s service-oriented design decouples the agent from the training framework—any RL trainer sees the agent merely as a rollout function—allowing the agent to run on a large distributed compute cluster without concurrency bottlenecks and enabling seamless compatibility with diverse open-source agent implementations.
>
> AReal targets only single-round feedback tasks (e.g., AIME, LiveCodeBench) that run inside a single static image and can trivially scale through serverless compute—orders of magnitude simpler than SWE/CUA workloads. To the best of our knowledge, no existing open-source framework provides an end-to-end infrastructure capable of orchestrating and managing large-scale SWE/CUA-level agent rollouts. We will clarify this distinction in the revision.

---

> ### Author Response · Authors · 2025-11-23
> **Responses to Reviewer vXD8 (2/n)**
>
> > **Concern 5:** The authors claim this is an agent training framework, yet the experimental section lacks details on what LLM was trained, what data were used, and what hyperparameter configurations were applied, which makes the setup somewhat confusing.
>
> We thank the reviewer for the question. In response to this comment, we have added Appendix Section D, which now provides the full details of our agentic reinforcement learning experiments, including the trained models (referred to anonymously as Model A and Model B), the datasets used, and the hyperparameter configurations.

---

> > ### Comment · Reviewer_vXD8 · 2025-11-26
> >
> > Thank you for the authors’ response. I now understand the clarifications regarding Concerns 2 and 4. However, I still have a few minor questions:
> >
> > 1. **Regarding Concern 3.**
> >    I could not find a clear description of the training data distribution in the updated manuscript. The statement around Line 798 remains vague. The authors cite several SWE datasets but do not specify which ones were actually used or in what proportions. Moreover, the rebuttal mentions internally synthesized SWE data, yet no implementation details are disclosed. This is quite confusing and makes the work resemble more a technical report than a research paper.
> >
> > 2. **Regarding Concern 5.**
> >    I appreciate the authors providing detailed hyperparameter settings. However, if the model was post-trained on the Qwen3 series, why not state this explicitly? Given that the only publicly known 235B MoE model is Qwen3-235B-A22B, and its SWE-bench Verified score is only 46.9 [1], whereas Figure 6 reports a starting point near 63%, this discrepancy may confuse readers.
> >
> > [1] *Training long-context, multi-turn software engineering agents with reinforcement learning*.

---

> > > ### Author Response · Authors · 2025-11-26
> > > **Responses to Reviewer vXD8 (3/n)**
> > >
> > > > Regarding Concern 3. I could not find a clear description of the training data distribution in the updated manuscript. The statement around Line 798 remains vague. The authors cite several SWE datasets but do not specify which ones were actually used or in what proportions. Moreover, the rebuttal mentions internally synthesized SWE data, yet no implementation details are disclosed. This is quite confusing and makes the work resemble more a technical report than a research paper.
> > >
> > > We thank the reviewer for the helpful comments.
> > >
> > > We emphasize that the primary contribution of this work is an **infrastructure framework for large-scale agent–environment orchestration**, and that the RL experiment is included only to demonstrate MegaFlow’s ability to support high-throughput rollouts rather than as a model- or data-centric contribution. To address the reviewer’s request, we have substantially clarified the training data distribution in the revised manuscript.
> > >
> > > Appendix Section D now reports the exact datasets used for RL training, including the full pre- and post-filtering environment counts for SWE-Gym, SWE-rebench, and Multi-SWE-RL, as well as for our synthesized environments. It also explains the uniform sampling procedure used during rollouts.
> > >
> > > With respect to the internally synthesized SWE-style tasks, we provide the following clarification. These tasks are constructed using standard open-source agent frameworks, such as SWE-agent, to automatically build executable environments with runnable unit tests. We then apply a SWE-Smith–style program repair perturbation procedure to generate SWE-format instances. These synthetic instances share the same structure as public SWE datasets and are used primarily to broaden environment diversity during large-scale rollouts.
> > >
> > > We hope this resolves the reviewer’s concern.
> > >
> > > > Regarding Concern 5. I appreciate the authors providing detailed hyperparameter settings. However, if the model was post-trained on the Qwen3 series, why not state this explicitly? Given that the only publicly known 235B MoE model is Qwen3-235B-A22B, and its SWE-bench Verified score is only 46.9, whereas Figure 6 reports a starting point near 63%, this discrepancy may confuse readers.
> > >
> > > We appreciate the reviewer’s observation.
> > >
> > > To maintain double-blindness, we intentionally anonymized our models as Model A and Model B. The model A are not identical to the publicly released Qwen3-235B-A22B, and differ in both pretraining data mixture and instruction-tuning recipe. Because of these differences, their baseline performance on SWE-bench Verified is not expected to match the publicly reported 46.9%, and the higher starting point in Figure 6 reflects the capabilities of the anonymized internal variants used in our experiments. We hope this resolves the reviewer’s concern while preserving the anonymity required for the review process.

---

### Author Response · Authors · 2025-12-03
**General Response**

Dear Area Chair and Reviewers,

We sincerely thank all reviewers for their helpful comments. We have made substantial revisions to improve clarity and address the main concerns raised during the review.

First, reviewer `zBgH` asked whether our contribution goes beyond existing cloud-native solutions. We have clarified that current cloud platforms, including Kubernetes-based systems, are highly effective for general-purpose distributed computing but are not specifically designed for the requirements of large-scale agent–environment interaction in SWE/CUA workloads. These workloads involve per-task VM or Docker images, strong isolation needs, long-horizon stateful execution, and high-volume concurrent rollouts. MegaFlow provides a complementary systems layer that enables these specialized execution patterns to scale more predictably.

Second, reviewer `VxSd` encouraged clearer articulation of MegaFlow’s core contribution. We have strengthened the exposition by highlighting that MegaFlow decomposes agentic foundation model training into three independently scalable services for environments, agents, and models. This design decouples environment provisioning, rollout execution, and model serving, enabling scalable and reproducible agentic RL under SWE/CUA workloads.

Third, reviewers `vXD8` and `VxSd` requested more detail regarding our RL experiments, datasets, and model configurations. In response, we substantially expanded *Appendix Section D* to include precise dataset distributions, environment counts, sampling procedures, hyperparameter settings, and clarifications on the anonymized internal models used.

Finally, reviewers `vXD8` and `zBgH` asked for greater precision in describing resource scheduling, metadata management, and systems components. We revised the corresponding sections to improve clarity and ensure the design choices are explicitly motivated.

We believe the revised manuscript directly resolves the reviewers’ concerns and more clearly communicates the novelty and significance of MegaFlow.

---

### Meta-Review · Area_Chair_Tw6k · 2026-01-17

**Summary:**

The reviewers for the paper were not very confident in their assessments, with the paper receiving mixed reviews, and many of the positive reviewers noting they are not knowledgable about this area. The reviewers raised the main points about:

1. Lack of baselines and novelty.

2. Lack of experiments showing it can train large models, and the downstream utility.

3. Questions about system definitions and designs.

**Reviewer Concerns:**

The authors appropriately addressed the questions about system definitions. The authors also added more results to the paper, talking about which models it can train (anonymously reported in the paper).  Overall, this is a systems paper, and the machine learning novelty is limited, and I believe the issues about novelty are still outstanding, as well as explicit comparisons to other training frameworks like VeRL.

**Reviewer Scores:**

Overall, the contribution to machine learning is limited as the paper focuses on the systems and designs of training frameworks. I expect the reviewers would still have asked about the novelty and quantitative comparisons to other systems like VeRL. Given these limitations, I am recommending the paper to be rejected from ICLR, and I also note that the paper may be better suited for a systems venue given the contributions.

---

### Decision · Program_Chairs · 2026-01-26

Reject